# Plasmochemical Modification of Crofer 22APU for Intermediate-Temperature Solid Oxide Fuel Cell Interconnects Using RF PA CVD Method

**DOI:** 10.3390/ma15124081

**Published:** 2022-06-08

**Authors:** Marta Januś, Karol Kyzioł, Stanisława Kluska, Witold Jastrzębski, Anna Adamczyk, Zbigniew Grzesik, Sławomir Zimowski, Marek Potoczek, Tomasz Brylewski

**Affiliations:** 1Faculty of Materials Science and Ceramics, AGH University of Science and Technology, A. Mickiewicza 30 Av., 30-059 Krakow, Poland; kyziol@agh.edu.pl (K.K.); kluska@agh.edu.pl (S.K.); witjas@agh.edu.pl (W.J.); aadamcz@agh.edu.pl (A.A.); grzesik@agh.edu.pl (Z.G.); brylew@agh.edu.pl (T.B.); 2Faculty of Mechanical Engineering and Robotics, AGH University of Science and Technology, A. Mickiewicza 30 Av., 30-059 Krakow, Poland; zimowski@agh.edu.pl; 3The Faculty of Chemistry, Rzeszow University of Technology, Powstańców Warszawy 12 Av., 35-959 Rzeszow, Poland; potoczek@prz.edu.pl

**Keywords:** Crofer 22APU, solid oxide fuel cell, interconnect, plasma modification, anti-wear layer

## Abstract

The results of plasmochemical modification on Crofer 22APU ferritic stainless steel with a SiC_x_N_y_:H layer, as well as the impact of these processes on the increase in usability of the steel as intermediate-temperature solid oxide fuel cell (IT-SOFC), interconnects, are presented in this work. The layer was obtained using Radio-Frequency Plasma-Activated Chemical Vapor Deposition (RF PA CVD, 13.56 MHz) with or without the N^+^ ion modification process of the steel surface. To determine the impact of the surface modification on the steel’s resistance to high-temperature corrosion and on its mechanical properties, the chemical composition, atomic structure, and microstructure were investigated by means of IR spectroscopy, X-ray diffraction (XRD), scanning electron microscopy (SEM) and energy dispersive spectroscopy (EDS). Microhardness, Young’s modulus, wear rate, as well as electrical resistance, were also determined. Micromechanical experiments showed that the plasmochemical modification has a positive influence on the surface hardness and Young’s modulus of the investigated samples. High-temperature oxidation studies performed for the samples indicate that N^+^ ion modification prior to the deposition of the SiC_x_N_y_:H layer improves the corrosion resistance of Crofer 22APU steel modified via CVD. The area-specific resistance of the studied samples was 0.01 Ω·cm^2^, which is lower than that of bare steel after 500 h of oxidation at 1073 K. It was demonstrated that the deposition of the SiC_x_N_y_:H layer preceded by N^+^ ion modification yields the best properties.

## 1. Introduction

In recent decades the circumstances related to the generation of energy have been getting increasingly critical. It is now impossible to satisfy human demand for energy using conventional energy sources only. Consequently, the search for new energy sources has become one of the priorities for the future development of human civilization. One of the potential alternatives to conventional energy conversion systems is the development of fuel cells. A fuel cell is an environmentally friendly device that generates electrical energy via the oxidation of fuel. This allows for the direct conversion of chemical energy into electrical energy without any intermediate conversion to thermal or mechanical energy. A single solid oxide fuel cell (SOFC) consists of a solid oxide electrolyte as well as two electrodes—the cathode and the anode, which are placed on the two sides of the electrolyte. To connect such cells in series and form a stack, bipolar plates known as interconnects are applied. Their purpose is to provide structural support in the cell, as well as a gas-tight separation between the cathode and anode spaces, and to ensure that the current flow through adjacent cells and can thus be supplied to receivers [1,2].

Over the last two decades, research has been conducted on the technology used to manufacture so-called intermediate-temperature solid oxide fuel cells (IT-SOFC). This type of fuel cell is capable of highly efficient operation at relatively low temperatures up to 873 K. Given this temperature range, the interconnect casing and plates can be produced using heat-resistant ferritic stainless steels (FSS), which are far cheaper than ceramic materials and metallic alloys based on nickel and cobalt [2,3,4,5,6].

Crofer 22APU, manufactured by ThyssenKrupp VDM [7,8], is an example of a metallic material designed specifically for application in IT-SOFC interconnects. Studies have shown that a dual-layer scale with a thicker Cr_2_O_3_ inner layer and a thinner outer layer composed of the MnCr_2_O_4_ spinel forms on this steel during oxidation. The outer layer inhibits the formation of volatile chromium oxides and oxy-hydroxides, which increase the activation polarization of the cathode [2]. The main issue stemming from the use of this steel is that its area-specific resistance (ASR) gradually increases together with the thickness of the oxide scale [2,3,4,5,8,9]. An overly high electrical resistance in the scale layer can significantly bottleneck the flow of electrical current from the cathode to the interconnect, thereby reducing the power output of the entire fuel cell stack [2]. The growth rate of the Cr_2_O_3_ layer thus has a considerable effect on the internal resistance of an IT-SOFC—a parameter that should be kept at the lowest possible level throughout the operating lifetime of the fuel cell. In recent years, many researchers have carried out experiments in order to decrease the high-temperature corrosion rate of interconnects [10,11,12,13,14,15], as well as improve their electrical [6,14,16,17,18,19,20,21,22,23,24,25] and mechanical properties [2,14]. The application of non-conventional, surface engineering methods, including plasma techniques, has yielded promising results.

During the last decades, amorphous hydrogenated carbon layers (a-C:H) obtained using chemical or physical vapor deposition methods [26,27] have been the target of extensive studies aimed at improving their unique mechanical, corrosion, and anti-wear properties [28]. Furthermore, their physicochemical parameters can already be initially specified during the process of designing their structure by means of, inter alia, the variable ratio of sp^3^/sp^2^ carbon bonds. This possibility, along with the opportunity of doping the structure of the aforementioned layers with atoms, e.g., Si, O, N, means that they have applications in many key branches. Therefore, to surface modification of the metallic interconnectors, doped carbon-based layers, especially SiCN:H layers, seem to be a promising alternative. One of the most promising and economically viable ways in which this objective can be achieved is the use of modern engineering methods on the surface properties of materials. One such method, and the most promising, is Plasma-Assisted Chemical Vapor Deposition (PA CVD) [29,30,31].

In these studies, we have used the RF (Radio Frequency) PA CVD technique for surface modification (plasma etching and/or N^+^ ion modification) and deposition of a SiC_x_N_y_:H layer on Crofer 22APU stainless steel. The next step was to investigate the influence of such a modification on the functional properties of the material (resistance to high-temperature corrosion, as well as its mechanical, tribological, and electrical properties).

## 2. Experimental Procedure

### 2.1. Material Preparation

The investigated material was Crofer 22APU ferritic stainless steel (ThyssenKrupp VDM GmbH, (Essen, Germany) with the following composition (wt.%): Cr-22.2, Mn-0.46, Si-0.03, Ti-0.06, Al-0.02, Ni-0.02, La-0.07 and Fe-balance [7]. Samples with dimensions of 10 × 20 × 1 mm were cut from a sheet of the investigated steel, ground with 100–1200-grit SiC abrasive papers, and then ultrasonically degreased for 20 min in isopropyl alcohol.

Then, oxides were removed from all the steel surfaces by means of etching using Ar^+^ ions (plasma conditions). In addition, a selected series was surface modified with plasma N^+^ ions. Both of these processes were carried out in plasma conditions using the Radio Frequency Chemical Vapour Deposition method (RF, 13.56 MHz, Elettrorava S.p.A, Venaria Reale, Italy).

In the subsequent stage, the RF reactor was used for hydrogenated silicon carbonitride layer deposition on steel samples (both sides), with (N^+^/SiC_x_N_y_:H series) or without the N^+^ ion modification process (SiC_x_N_y_:H series). The layer was obtained using a reactive gas mixture consisting of silane (SiH_4_), methane (CH_4_), and nitrogen (N_2_). The technological parameters for the applied plasmochemical processes are presented in Table 1.

### 2.2. Research Methods

The morphology and chemical composition of the tested samples were investigated by means of scanning electron microscopy (SEM) performed using an FEI Nova NanoSEM 200 microscope coupled with an EDAX Genesis XM2 system required for the analysis of chemical composition via energy-dispersive X-ray spectroscopy (EDS).

The phase compositions of the samples were studied by means of X-ray diffraction (XRD) performed with a Philips Analytical X’Pert PW 3710 diffractometer, using monochromatic CuKa radiation. The qualitative phase composition of the prepared materials was established by comparing the obtained diffraction patterns with data included in the ICSD database.

The atomic structure of the obtained layers was determined by means of FTIR spectroscopy with the use of the Bruker Vertex 70V vacuum spectrometer (resolution of 4 cm^−1^) with a module dedicated to the registration of absorption and reflectance spectra. Measurements were performed for the SiC_x_N_y_:H layer obtained on a (001) Si substrate and steel. In the first case, the spectra were obtained using the transmission technique, while in the case of steel, reflective techniques were used.

The microgeometry of the surface of the investigated materials was determined using a Hommel T500 surface profiler manufactured by Hommelwerke GmbH (Wiesbaden, Germany).

Microindentation hardness (H_IT_) and Young’s modulus (E_IT_) were evaluated by means of an instrumental indentation method with the use of the Micro–Combi–Tester (MCT) manufactured by CSM Instruments. These measurements were performed using a diamond indenter (Berkovich) with a triangular-shaped tip and half angle of 65.35° for three different applied loads: 10, 100, and 500 mN.

A scratch test was performed in compliance with the PN-EN 1071-3 standard, using a Rockwell graves diamond indenter with a tip radius of 200 µm. The maximum load on the diamond stylus was 10 N (P_max_), and it increased linearly from 0.01 N to P_max_ along the entire length of the scratch track, which was 3 mm. The rate at which the indenter moved in relation to the sample was 3 mm·min^−1^. As a result of the scratch tests, the critical load L_C1_, defined as the load associated with cohesive cracks in the layer, and L_C2_, which causes adhesive failure of the layer-substrate system, were determined. The different kinds of wear were determined on the basis of microscopic observations of the scratch track and the analysis of the friction force, penetration depth, and acoustic emission measured during the scratch test.

The wear resistance of the layers was tested during dry friction using the ball-on-disc method. The measurements were performed at a load (F_n_) of either 0.5 N or 1 N, a rotational speed of 60 rpm (revolution per minute), friction radius of 3 mm, and a maximum of 1500 cycles (revolutions) for layers as well as 2000 cycles for steel substrate. The sliding counter element was an Al_2_O_3_ ball with a 6 mm diameter.

The evaluated parameters included the friction coefficient, determined from the ratio of the respective forces:
(1)
$$\mu =\frac{F_{t}}{F_{n}}$$

where: F_t_—tangential force [N], and F_n_—normal force [N].

The wear rate was calculated using the following formula:
(2)
$$W_{V}=\frac{V}{F_{n}\cdot s}$$

where: V—volume of the worn material [mm^3^], calculated from the mean surface area of the groove cross-section, F_n_—normal force [N], and s—sliding distance [m].

The oxidation kinetics of unmodified steel and the steel samples with the deposited layer were investigated under isothermal conditions, for up to 500 h, in laboratory air at 1073 K, using a thermogravimetric apparatus with 10^−7^ g sensitivity. The applied high-temperature thermogravimetric apparatus has been described previously [32].

The electrical resistance of the samples was measured by means of the four-point two-probe technique under a constant current density of 0.1 A·cm^2^ in the air; over the range of 623–1073 K. Prior to the measurements, the samples were pre-oxidized for 500 h in the air at 1073 K. The potential drop for each sample was measured by means of an HP digital multimeter (34401) with 0.3% precision. The apparatus used to measure the electrical resistance and the sample preparation procedure are described in [33].

The electrical resistance of an oxidized specimen is usually measured in terms of its area-specific resistance (ASR), which is defined as the product of resistance and the contact surface area of the oxide and the steel. Due to the symmetrical design of the sample, the area-specific resistance of the samples was calculated on the basis of the obtained resistance values using the following formula:
(3)
$$ASR=\frac{R\cdot A}{2}$$

where: R—electrical resistance [Ω], and A—surface area of the Pt layer [cm^2^].

## 3. Results and Discussion

### 3.1. Chemical Composition and Surface Parameters

Figure 1 illustrates SEM surface images of Crofer 22APU steel, both unmodified and after depositing layers, without (SiC_x_N_y_:H series) and with prior plasma N^+^ ion modification (N^+^/SiC_x_N_y_:H series).

Several delaminations can be observed on the unmodified steel surface due to the rolling of the steel sheet and scratches resulting from grinding. The silicon carbonitride layer deposited on the steel samples without N^+^ ion modification (Figure 1b) and after N^+^ ion modification (Figure 1c) does not significantly reduce the visibility of these surface defects.

EDS results obtained from the surface of the tested samples confirm the approximate layer chemical compositions, determined as SiC_1.4_N_0.8_ in the case of the sample without N^+^ ion modification and SiC_1.5_N_0.9_ for the N^+^ ion modified material. In addition, combined SEM-EDS cross-section measurements revealed that layers are homogenous in terms of chemical composition. The below images (Figure 2a,b) demonstrate SEM cross-sections of steel samples from both series (with the SiC_x_N_y_:H layer and with the previously nitrided surface and the same layer applied). In the first case, the average layer thickness does not exceed 1.0 µm and is about 1.7 µm for the nitrided series.

The parameters of the two-dimensional profile were determined based on the vertical deviation from the line of reference. During these examinations, the following values were compared: the arithmetic mean of the level of roughness (R_a_), the maximum difference between the heights of the highest peak and the lowest valley (R_z_), and the arithmetic mean of the heights of the five highest peaks and the five deepest valleys in the roughness profile (R_t_). The values determined from the roughness test performed on individual samples are presented in Table 2.

The results of the roughness test show a decrease in all parameters for the substrates with the SiC_x_N_y_:H layer preceded by the N^+^ ion modification process (to a large extent, the R_z_ and R_t_ parameters).

Figure 3a,b compiled the FT IR spectra of the layers obtained on the (001)Si and Crofer 22APU substrates without N^+^ ion modification and those subjected to the previous N^+^ ion modification process.

The spectra of the layers on (001)Si were recorded using the transmission technique, whereas the layers on steel were determined by means of the reflection technique. All spectra are dominated by a wide, intense band with a complex envelope in the range of 600–1200 cm^−1^ with a maximum of 880 cm^−1^ for a layer on (001) Si and ca. 755 cm^−1^ in the case of a layer on Crofer 22APU. Apart from the above-mentioned bands, in the spectra range of ca. 1250–1600 cm^−1^, there are weak bands at the locations of ca. 1570 cm^−1^ (=NH, -NH_2_), ca. 2155 cm^−1^ (Si-H, C ≡ C, C = N), centered ca. 2890 cm^−1^ (CH_n_, *n* = 1,2,3) and ca. 3370 cm^−1^ (-NH_2_, =NH, -OH).

Despite the use of various FT IR techniques (transmission and reflection), the range of these bands and their maxima (in this range) coincide with the layers obtained on the applied substrates. The intensities of the layer strips preceded by N^+^ ion modification do not differ much from those without N^+^ ion modification, which suggests that the earlier surface treatment does not have a significant effect on their growth rate.

The data in Figure 4a,b demonstrates that a strong, complex band in the range of ca. 600 cm^−1^ to ca. 1250 cm^−1^ is most important for describing the structure of the obtained layers. The intensity and shape of this band are complex for the substrates used. There can be two reasons for this fact—the use of different FT IR measurement techniques (transmission, reflection) or the different formation rates of appropriate groups during the preparation process. The first seems unlikely due to the fairly good compatibility of the band positions s determined by both techniques in the higher frequency range from 1250 cm^−1^ to 4000 cm^−1^.

Interpretation of such complex spectra was based on the deconvolution of the band in the range 400–1300 cm^−1^ using the program “Fityk 0.9.8” [34], as a result of which the component bands were obtained—Figure 4a–d.

Small differences in the positions of these bands are understandable and result from the use of two different measuring techniques. In the spectra of layers obtained on the (001) Si substrate (Figure 3a,b), the band resulting from Si-N stretching vibrations (930 cm^−1^) has the highest intensity, followed by Si-C-N (stretching), Si-CH_3_ (bending, 810 cm^−1^), Si-CH_2_ (bending), Si-O-Si (stretching, 1035 cm^−1^), =NH (bending, 1160 cm^−1^), Si-C (stretching, 700 cm^−1^) and very weak Si-CH_3_—(stretching, 1260 cm^−1^). The mutual proportions of the aforementioned bands are different in the case of layers obtained on the steel substrate. First of all, the intensities of the bands originating from vibrations in Si-N and =NH groups decrease drastically, and the intensities of the SiC-C, Si-(CH_3_)_n_ bands coincide with the band determined in groups where silicon is bound to nitrogen and carbon (Si-C-N).

In chemical vapor deposition technologies, the influence of the type of substrate on the growth mechanism of thin films and layers is often observed. The substrate is catalytic [35]. In this case, the (001) Si substrate promotes the growth of layers with a predominance of Si-N bonds, while layers on the steel consist of silicon carbide, hydrogenated silicon carbide, and silicon carbonitride.

The proportion of individual atoms forming the layers varies depending on the preparation of the substrate. These changes are visible by comparing the spectra in Figure 4a–d and the data in Table 3 obtained by comparing the areas covered by the individual bands.

In the case of the (001) Si substrate, the N^+^ ion modification process promotes the formation of Si-C-N bonds and hydrogenated silicon carbide Si-(CH_3_)_n_, Si-CH_2_-Si at the expense of Si-C, Si-N, N-H, Si-CH_3_ bonds. In the layers obtained on steel substrates (Figure 4c,d), the band in the range of 700–950 cm^−1^ with a maximum of 815 cm^−1^ visible in the spectra of layers on monocrystalline silicon (Figure 4a,b) was separated into Si-C-N and Si-(CH_3_)_n_. The N^+^ ion modification process of this substrate does not change the share of Si-C-N bonds but leads to an increase in the share of Si-(CH_3_)_n_, N-H, and Si-CH_3_ bonds.

In summary, it can be stated that SiC_x_N_y_:H layers obtained on various substrates in the RF CVD process have a complex chemical composition, containing silicon, carbon, nitrogen, and hydrogen in various groups. These groups are present in all layers, but their proportions depend on the type of substrate surface.

### 3.2. Mechanical and Tribological Properties

The hardness and elastic modulus are important indexes of the mechanical properties of thin films and have an important impact on tribological performance. Measured values of hardness and Young’s modulus were presented in Table 4 for all series of samples: the unmodified Crofer 22APU steel, the steel without N^+^ ion modification before the SiC_x_N_y_:H layer deposition (SiC_x_N_y_:H series), and the N^+^ ion modified steel with the SiC_x_N_y_:H layer (N^+^/SiC_x_N_y_:H series).

When compared with the unmodified Crofer 22APU sample, both the microindentation hardness and elasticity modulus of samples after plasma N^+^ ion modification and plasma layer deposition are significantly improved.

The measurement values of surface hardness observed for low loads on both samples with deposited layers (without or with N^+^ ion modification) are around five and four times higher, respectively. An increase in the load causes this difference to diminish. Such changes are typical of layer/substrate interfaces since, at a given load (i.e., the depth of the indenter penetration), the continuity of the layer is broken. As with H_IT_, the values of Young’s modulus (E_IT_) are higher for the steel samples with deposited layers (Table 4). The value of Young’s modulus and hardness of SiC_x_N_y_:H layers on N^+^ ion modification of Crofer 22APU steel are much higher than those obtained by the Authors [36,37,38]. At a load of 10 mN, the highest E_IT_ value is observed for the steel with N^+^ ion modification and the deposited layer. These differences can be explained on the basis of FT IR studies (see Figure 4). FT IR analysis shows that SiC_x_N_y_:H layers deposited on N^+^ ion modified substrates have a higher content of Si-C bonds than the layers deposited on the steel without N^+^ ion modification. Moreover, in the case of the steel after N^+^ ion modification and layer deposition, Young’s modulus values are higher. This may be associated with the modification of the surface using N^+^ ions and the atomic structure of the layers, namely the higher contribution of the Si-C bonds in the steel sample after N^+^ ion modification and Young’s modulus values that correspond to individual phases (310 GPa for crystalline Si_3_N_4_ and 450 GPa for crystalline SiC [39]).

Studies of the layer damage and penetration depth at the progressive loading of the Rockwell indenter showed that the obtained layers demonstrate comparable scratch resistance. The observed cracks were curved in a direction opposite to the motion of the indenter. This leads to the conclusion that these cracks were caused by tensile stress that develops behind the indenter. The number of cracks increased when the load exerted by the indenter increased. When a critical load L_C1_ of around 3.5 N was applied in the experimental series with the SiC_x_N_y_:H layer but without N^+^ ion modification, the cracks formed a network that arose across the entire width of the scratch. In the case of the samples after N^+^ ion modification, similar damage was observed; however, a load of 3 N was applied. Adhesive failure was observed when applying a critical load L_C2_ of 9 N—in the case of the SiC_x_N_y_:H sample and 7 N for the N^+^/SiC_x_N_y_:H sample. It can be concluded that without N^+^ ion modification of the steel surface prior to the deposition of the SiC_x_N_y_:H layer, higher scratch resistance is achieved. The obtained SiC_x_N_y_:H/substrate system also exhibits higher stiffness in comparison to the system obtained with prior N^+^ ion modification. The maximum penetration depths were 13 µm and 15 µm, respectively. Images of the scratch track during the scratch test are shown in Figure 5a–f.

During sliding contact with the A_2_O_3_ ball, the coefficient of friction (COF), compared with the unmodified Crofer 22APU surface (0.2 before and 0.85 after 300 cycles) of two experimental series (SiC_x_N_y_:H and N^+^/SiC_x_N_y_:H) of samples was in the range of 0.2–0.4 when the layers were continuous. Friction was found to increase until the total wear of the layer, and then COF achieved a value around 0.8, similar to that determined for the uncoated substrate. In the case of the N^+^/SiC_x_N_y_:H samples, this occurred after around 60 cycles at a load of 1 N and 180 cycles at a load of 0.5 N. Whereas for the SiC_x_N_y_:H series, this process was not as rapid. Instead, the layer was successively abraded; significant wear was only observed after ca. 800 cycles, and total wear only occurred after 1500 cycles at a load of 1 N.

Table 5 presents values of the wear rate and grove depth of tested samples after friction tests. The series with the SiC_x_N_y_:H layer deposited after N^+^ ion modification exhibits a much lower wear resistance compared to the series modified exclusively via SiC_x_N_y_:H layer deposition. In the former case, the wear rate is as high as 503 mm^3^·N^−1^·m^−1^, while for the SiC_x_N_y_:H series, it is equal to 71 mm^3^·N^−1^·m^−1^.

Images of the wear track of the samples after the ball-on-disc test are shown in Figure 6. The wear of both layers was mainly abrasive and additionally intensified by cracking (Figure 6a,b). The comparable value of the wear rate of the unmodified Crofer 22 APU steel is the result of the seizing process and the smoothing of the surface roughness due to the filling of the scratches by the wear products (Figure 6c). This finding for the frictional wear of unmodified steel is also confirmed in the scratch test (Figure 5e,f).

### 3.3. Corrosion Study

One of the most important requirements for exposing metallic interconnects to oxidizing SOFC atmospheres at 1073 K is good resistance against high-temperature corrosion. In order to compare the properties of the Crofer 22APU steel with the silicon carbonitride layer deposited using RF CVD without N^+^ ion modification (SiC_x_N_y_:H) and after prior N^+^ ion modification (N^+^/SiC_x_N_y_:H) and those of unmodified steel—all after oxidation in air at 1073 K—their oxidation kinetics were investigated over a period of 500 h using isothermal thermogravimetric analysis.

Figure 7 shows the oxidation kinetics of these three types of samples, expressed by a function of mass gain per surface area unit vs. time.

The presented kinetics curves are not adjusted for the formation of volatile chromium compounds, which was negligible. When the latter effect is not accounted for, the relative error is below 3% as long as the oxidation time does not exceed 1200 h.

The growth of scales and/or the intermediate reaction layer is usually quantified by determining the mass of oxygen bound per unit of the metal surface area. The Pilling–Bedworth equation was therefore used to plot the kinetics curve and subsequently calculate the parabolic rate constant of oxidation (k_p_) [40]:
(4)
$${\left(\frac{\Delta m}{A}\right)}^{2}=k_{p}\cdot t+C_{\left(y\right)}$$

where: ∆m/A—mass gain per unit area [g·cm^−2^], k_p_—parabolic rate constant [g^2^·cm^−4^·s^−1^], t—reaction time [s], and C—integration constant defining the onset of parabolic kinetics.

The lowest corrosion rate and subsequently the highest resistance to high-temperature corrosion in the air were observed in the case of the steel with N^+^ ion modification and the deposited layer (N^+^/SiC_x_N_y_:H). Of all the investigated samples, the lowest corrosion resistance was exhibited by the steel without N^+^ ion modification and with the deposited layer (SiC_x_N_y_:H). In addition, it should be noted that the corrosion rate of the steel with N^+^ ion modification and the deposited layer (N^+^/SiC_x_N_y_:H) in the time interval of 0–230 h is higher than that observed for unmodified steel, subsequently decreasing in a systematic manner beyond that time. This is due to the fact that unmodified steel starts to corrode at a different rate after 230 h.

The experimental kinetics data obtained in this study, including the parabolic rate constant (k_p_) of the oxide product growth—determined from Equation (4) using regression analysis—and the time intervals (t) in which the parabolic rate law is obeyed are collected in Table 6.

The comparison of these data shows that the parabolic rate constant (k_p_) for the N^+^ ion modified steel with the deposited layer (N^+^/SiC_x_N_y_:H) was about half an order of magnitude lower than the parabolic rate constant measured for the steel without N^+^ ion modification and with the deposited layer (SiC_x_N_y_:H). The observed oxidation behavior demonstrated the thermally protective role of the N^+^/SiC_x_N_y_:H layer in air at 1073 K.

It is worth comparing the values given in Table 6 with those achieved using Mn-Co-O spinel layers deposited as protective-conducting coatings on Crofer 22APU steel. For example, one study reported a k_p_ value of 3.90 × 10^−14^ g^2^·cm^−4^·s^−1^ for a Crofer 22APU sample with an MnCo_2_O_4_ coating [41], while in another, an Mn_1.5_Co_1.5_O_4_ coating likewise deposited on this type of steel was found to have a k_p_ of 1.45 × 10^−14^ g^2^·cm^−4^·s^−1^ [42]. In both studies, the samples had been oxidized at 1073 K for 1000 h. Although these values are slightly lower than those determined for the samples studied in this work, application of the Mn-Co-O spinel coating does not merely reduce the corrosion rate but also inhibits chromium evaporation, which is the cause of cathode poisoning [43]. Therefore, it is to be expected that applying this type of ceramic coating on the Crofer 22APU/Ar^+^/N^+^/SiC_x_N_y_:H layered system would reduce its oxidation rate even further.

XRD studies (Figure 8) revealed that Cr_2_O_3_, MnCr_2_O_4_, and TiO_2_ phases formed during prolonged exposure of the unmodified steel to the corrosive conditions. In the case of the steel sample without N^+^ ion modification, Mn_2_O_3_ was detected along with the other previously mentioned phases, whereas only MnCr_2_O_4_ and Cr_2_O_3_ were determined on the N^+^ ion modified sample. In addition, there are high-intensity reflections that can be attributed to the α-Fe phase that constitutes the metallic substrate in every diffraction pattern presented in this figure.

Figure 9 shows the polished cross-sections of corrosion products formed on each of the three types of samples after 500 h of oxidation in air at 1073 K, as well as EDS point analyses at the locations indicated using numbers 1–6 in Figure 9c.

In the case of the unmodified steel, a two-layer scale was grown, which contained a thick and continuous Cr_2_O_3_ inner layer and a thinner outer layer built of MnCr_2_O_4_. The scale was compact and well-adherent to the metallic substrate. The complete scale thickness varied; however, its average value could be estimated at around 3.5 µm (Figure 9a). On the other hand, a sublayer with the same chemical composition as the substrate with the SiO_2_ addition could be seen in the two modified steel samples (Figure 9c, point ‘5’), and the scale grown above this sublayer contained Cr_2_O_3_ (Figure 9c, point ‘4’) and MnCr_2_O_4_ (Figure 9c, points ‘1’ and ‘2’).

EDS point analyses showed that the corrosion product also consists of amorphous SiO_2_ phase precipitates (Figure 9c, point ‘3’). In the case of the modified steel without N^+^ ions (SiC_x_N_y_:H), the previously described ca. 1.8 µm thick sublayer gradually transforms into chromia and the manganese–chromite spinel as a result of interdiffusion between the steel and layer components. Consequently, a considerable increase in the scale thickness increases significantly in accordance with the oxidation kinetics measurements performed on the sample. Conversely, the well-adherent compact ca. 1.2 µm thick sublayer, grown on the steel with N^+^ ion modification and the deposited layer (N^+^/SiC_x_N_y_:H), effectively hinders interdiffusion, thereby increasing corrosion resistance.

### 3.4. Electrical Properties

From a practical perspective, it is essential that modifications aimed at decreasing the rate of corrosion do not cause a simultaneous increase in the electrical resistance of the oxidation product. Such an increase may be caused by numerous factors, such as a partial loss of contact with the metallic substrate, changes in the structure and phase composition of the scale, and changes in the microstructure of the scale and/or the scale/steel interface. To address this issue, the electrical resistance of the pre-oxidized steel without N^+^ ion modification and with the deposited layer (SiC_x_N_y_:H) and the steel after N^+^ ion modification with the deposited layer (N^+^/SiC_x_N_y_:H) was measured and compared with that of unmodified steel.

Temperature dependences of area-specific resistance (ASR) for the unmodified Crofer 22APU steel, steel without N^+^ ion modification and with the deposited layer (SiC_x_N_y_:H), and steel after N^+^ ion modification with the deposited layer (N^+^/SiC_x_N_y_:H) after 500 h of oxidation in air at 1073 K are presented in the form of Arrhenius plots in Figure 10.

The plots show that the logarithm of ASR increases linearly with decreasing temperature. This trend is typical for semiconductors, including chromia. The linearity of the dependences shown in Figure 10 indicates the temperature-activated nature of electrical conduction in the investigated steel/scale and steel/layer systems, as expressed by the equation [3]:
(5)
$$ASR=\left(\frac{A}{T}\right)\exp\left(\frac{E_{c}}{kT}\right)$$

where: A—pre-exponential factor [Ω·cm^−2^·K^−1^], Ec—activation energy [kJ·mol^−1^], k—Boltzmann constant [eV·K^−1^], and T—absolute temperature.

Table 7 lists the area-specific resistance (ASR) values of the studied Crofer 22APU/scale, Crofer 22APU/SiC_x_N_y_:H, and Crofer 22APU/N^+^/SiC_x_N_y_:H layered systems oxidized in air at 1073 K as well as the values of activation energy of electrical conduction (E_c_) for the entire investigated temperature range, determined from the linear dependence ln(ASR) = f(1/T).

The values of activation energy of electrical conduction observed for the N^+^ ion modified steel with the deposited layer (N^+^/SiC_x_N_y_:H) and steel without N^+^ ion modification with the deposited layer (SiC_x_N_y_:H) are comparable to the value determined for oxide scales formed on unmodified Crofer 22APU steel (Table 7). It is therefore likely that the transport of electron carriers within the investigated layered system is determined by the movement of small polarons in the crystal lattice of the chromia scale, which is the main oxidation product formed on the modified and un-modified Crofer 22APU steel.

In addition, the data shown in Table 7 also indicates that the ASR values of the two investigated samples with deposited layers measured at 1073 K are lower than that of unmodified Crofer 22APU with the chromia scale. The ASR value for the N^+^ ion modified steel with the deposited layer (N^+^/SiC_x_N_y_:H) is lower than that determined for the steel without N^+^ ion modification with the deposited layer (SiC_x_N_y_:H), which can be associated with better adhesion of the corrosion product to the substrate in the case of the former, as well as the absence of foreign phases in the layer adjoining the scale, which is observed in the steel without N^+^ ion modification and with the deposited layer (SiC_x_N_y_:H), Figure 10. The fact that unmodified Crofer 22APU steel exhibits the highest ASR of all samples is due to the considerable thickness of the scale, which is composed mostly of chromia; the electrical conductivity of chromia is lower than that of the manganese–chromium spinel [2]. It is worth mentioning that the ASR levels observed for the steel sample that had undergone modification with N^+^ ions and the deposited layer (N^+^/SiC_x_N_y_:H) do not exceed 0.1 Ω·cm^2^ in the temperature range of 773–1073 K, which is the threshold set for interconnect materials, whereas in the case of the sample with the deposited SiC_x_N_y_:H layer, but no N^+^ ion modification, the temperature range for which this was true is more narrow by 100 K (873–1073 K). The ASR measured for the steel sample with no modification at the highest measurement temperature (1073 K) is just below the threshold value set for IT-SOFC interconnects, and it exceeds this threshold at lower temperatures.

It can be concluded from the conducted study that the deposition of the SiC_x_N_y_:H layer on the Crofer 22APU ferritic steel preceded by nitriding significantly improves its electrical properties and may be used to increase SOFC operating time. Thus, in view of the expectations concerning fuel cell operating time, which are at a level of 40,000 h, the surface modification of steel may be of fundamental importance for fuel cell life.

## 4. Conclusions

The conducted research confirmed that the RF PA CVD technique is suitable for surface modification of Crofer 22APU steel, obtaining amorphous silicon carbonitride layers in the process. Additionally, we can conclude that:

N^+^ ion modification decreases the modified steel substrate wear resistance ca. 7 times compared to the value obtained from the material with only the SiC_x_N_y_:H layer.Micromechanical experiments demonstrated that plasmochemical modification significantly influences the mechanical properties (hardness and Young’s modulus) of the materials.Oxidation kinetics measurements performed for an oxidation time of 500 h and a temperature of 1073 K indicate that prior N^+^ ion modification in plasmochemical conditions results in improved corrosion resistance of the Crofer 22APU steel with the deposited SiC_x_N_y_:H layer.Structural, microstructural, and chemical composition studies performed by means of XRD and SEM-EDS revealed the presence of a dual-layer scale composed of chromia and the MnCr_2_O_4_ spinel as the outer layer.Based on electrical conductivity measurements of the Crofer 22APU (unmodified)/scale, Crofer 22APU/SiC_x_N_y_:H and Crofer 22APU/N^+^/SiC_x_N_y_:H layered systems in the temperature range of 623–1023 K in air, it was determined that the deposition of the ceramic layer on the surface of the Crofer 22APU improves the electrical properties of the studied ferritic steel.The steel sample surface-modified via N^+^ ion modification and the deposition of the SiC_x_N_y_:H layer and oxidized in the aforementioned conditions exhibited the lowest ASR value, i.e., 0.01 Ω·cm^2^.

All of the obtained results suggest that the silicon carbonitride layer deposition preceded by N^+^ ion modification can successfully improve the properties of Crofer 22APU steel, thereby increasing its potential to be used as an interconnect material in planar-type intermediate-temperature solid oxide fuel cells.

## Figures and Tables

**Figure 1 materials-15-04081-f001:**

SEM images of the surface of the Crofer 22APU steel: (**a**) unmodified, (**b**) after SiC_x_N_y_:H layer deposition, and (**c**) after plasma N^+^ ion modification process and SiC_x_N_y_:H layer deposition.

**Figure 2 materials-15-04081-f002:**
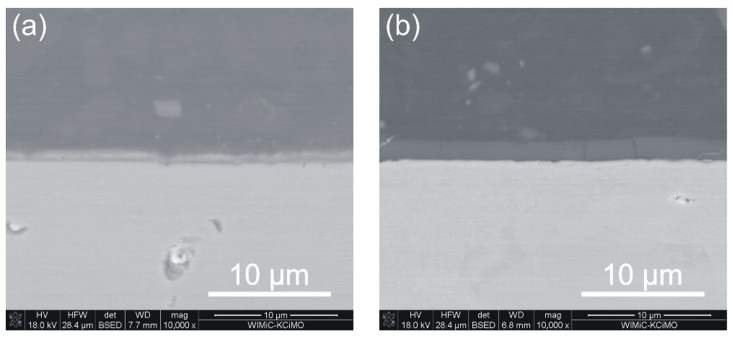
SEM images of the Crofer 22 APU steel samples cross-sections: (**a**) with SiC_x_N_y_:H layer, (**b**) with SiC_x_N_y_:H layer preceded by the nitriding process.

**Figure 3 materials-15-04081-f003:**
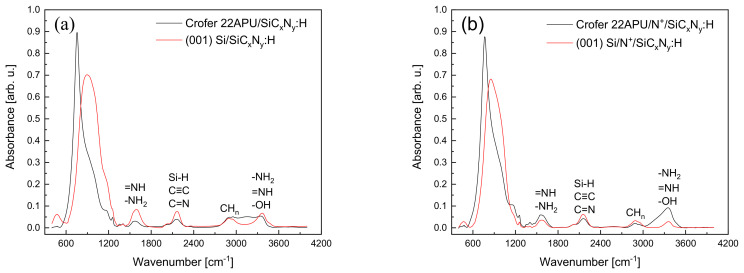
FTIR spectra of modified (001) Si and Crofer 22APU surfaces: (**a**) SiC_x_N_y_:H layer, (**b**) N^+^ ion modification, and SiC_x_N_y_:H layer.

**Figure 4 materials-15-04081-f004:**
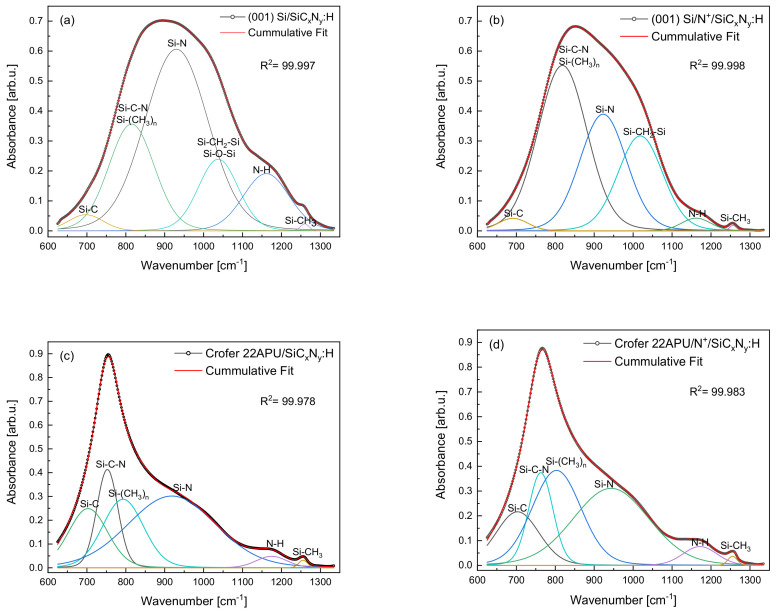
Deconvolution of FT IR spectra in the range of 600–1350 cm^−1^ for (**a**) SiC_x_N_y_:H layer on (001) Si, (**b**) N^+^ ion modification, and SiC_x_N_y_:H layer on (001) Si, (**c**) SiC_x_N_y_(H) layer on Crofer 22APU and (**d**) N^+^ ion modification and SiC_x_N_y_:H layer on Crofer 22APU.

**Figure 5 materials-15-04081-f005:**
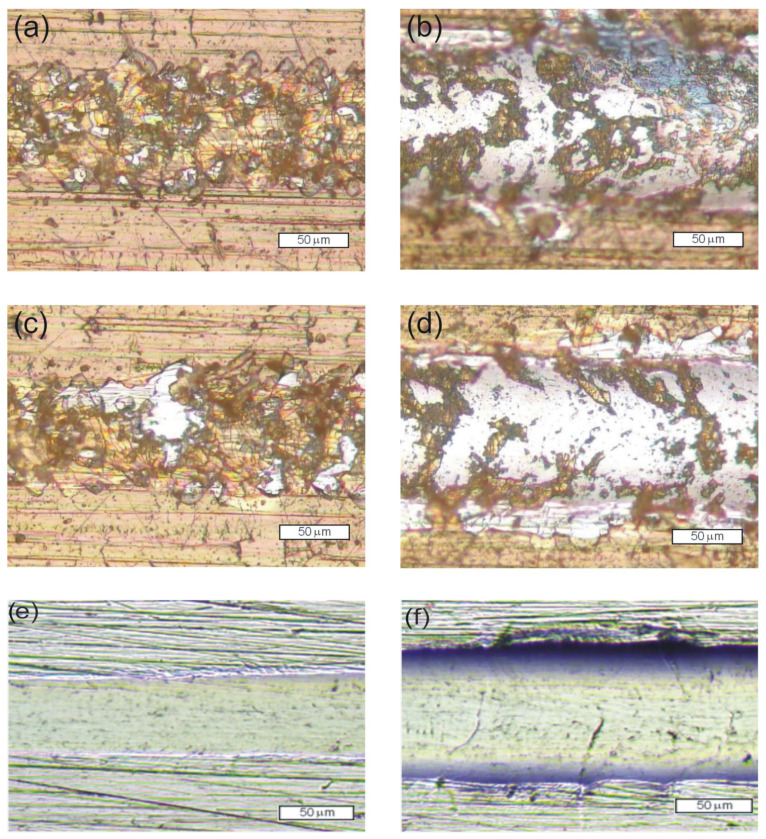
Images of the scratch track at locations where a characteristic damage of the layer occurred: (**a**) SiC_x_N_y_:H at L_C1_ = 3.4 N, (**b**) SiC_x_N_y_:H at L_C2_ = 8.9 N, (**c**) N^+^/SiC_x_N_y_:H at L_C1_ = 2.9 N and (**d**) N^+^/SiC_x_N_y_:H at L_C2_ = 7.3 N, (**e**) unmodified Crofer 22APU steel at 3 N, (**f**) unmodified Crofer 22APU steel at 8 N.

**Figure 6 materials-15-04081-f006:**
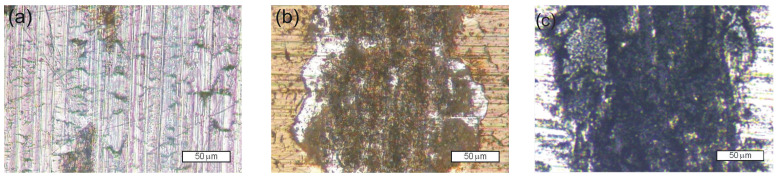
Images of the wear track of the samples after the ball-on-disc test: with SiC_x_N_y_:H layer,1 N-1500 cycles (**a**), after N^+^ ion modification and SiC_x_N_y_:H layer, 0.5 N-180 cycles (**b**) and unmodified Crofer 22APU steel (**c**).

**Figure 7 materials-15-04081-f007:**
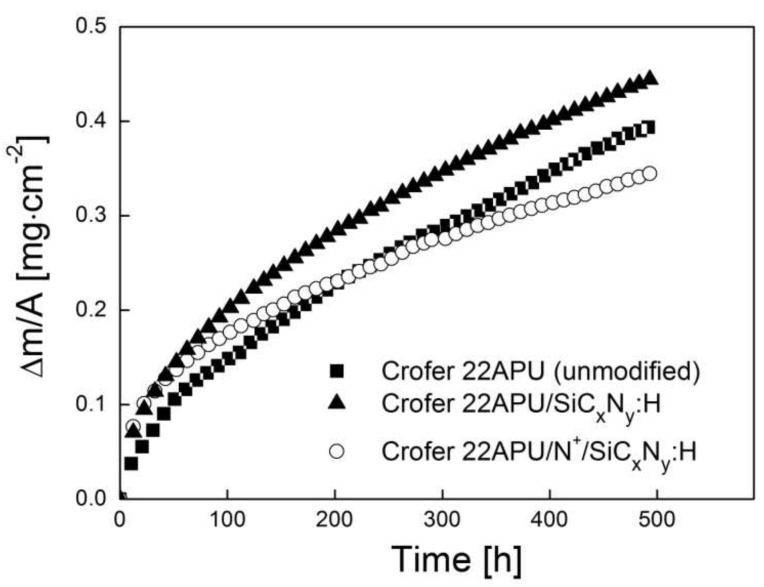
Oxidation kinetics of unmodified Crofer 22APU steel, steel with the deposited layer (SiC_x_N_y_:H) without prior N^+^ ion modification, and with N^+^ ion modification process and layer deposition (N^+^/SiC_x_N_y_:H) in the air at 1073 K in a linear system of coordinates.

**Figure 8 materials-15-04081-f008:**
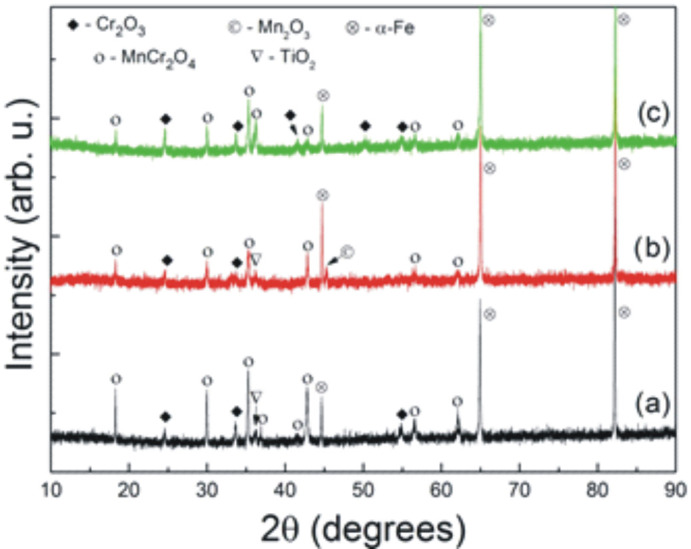
Diffraction patterns recorded for: (**a**) unmodified Crofer 22APU steel, (**b**) Crofer 22APU with the deposited SiC_x_N_y_:H layer, and (**c**) Crofer 22APU with the layer deposited after N^+^ ion modification (N^+^/SiC_x_N_y_:H) obtained after oxidation at 1073 K in the air for 500 h.

**Figure 9 materials-15-04081-f009:**
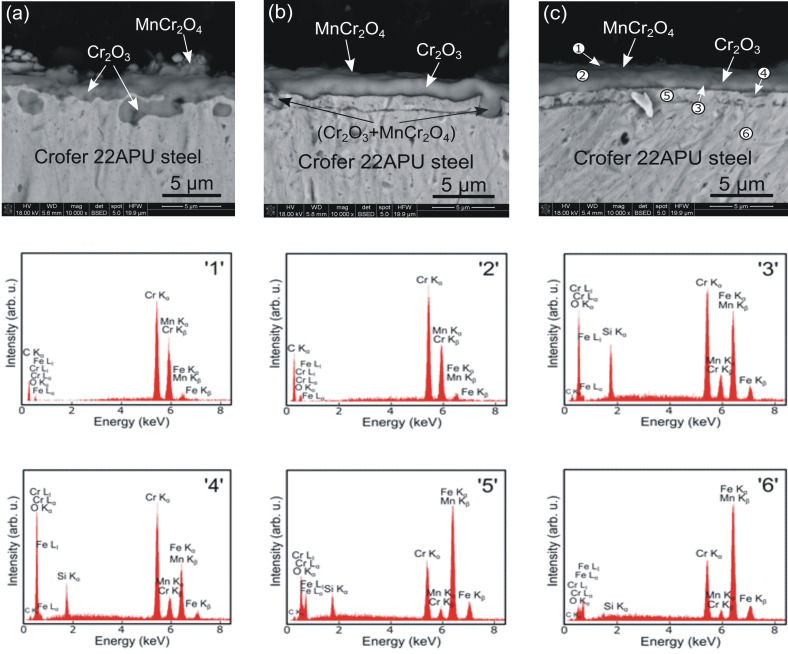
SEM cross-section microphotographs of (**a**) unmodified Crofer 22APU steel, (**b**) Crofer 22APU with the deposited SiC_x_N_y_:H layer, and (**c**) Crofer 22APU with the layer deposited after N^+^ ion modification (N^+^/SiC_x_N_y_:H) obtained after oxidation at 1073 K in the air for 500 h as well as EDS point analyses spectra obtained from locations ‘1’–‘6’ indicated in Figure 9c.

**Figure 10 materials-15-04081-f010:**
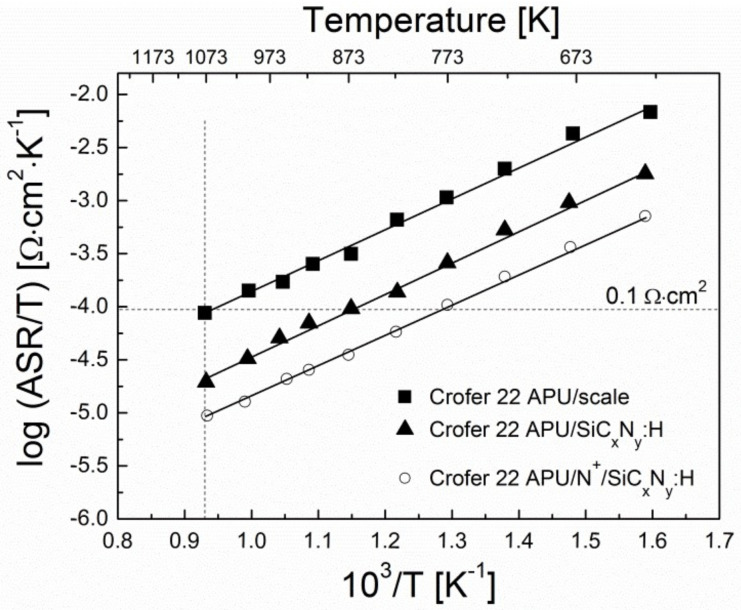
ASR vs. temperature dependency for the unmodified Crofer 22APU steel, steel without N^+^ ion modification and with the deposited layer (SiC_x_N_y_:H), and steel after N^+^ ion modification with the deposited layer (N^+^/SiC_x_N_y_:H) after 500 h of oxidation in air, in an Arrhenius plot.

**Table 1 materials-15-04081-t001:** Technological parameters chosen for Crofer 22APU steel surface modification by means of the RF CVD process.

Parameters	Ion Etching	N^+^ Ion Modification	SiC_x_N_y_:H Deposition
RF plasma density [W·cm^−2^]	0.6		
Gas precursor/flow rate [cm^3^·min^−1^]	Ar/100	N_2_/80 Ar/100	SiH_4_/3 CH_4_/10 N_2_/80 Ar/100
Pressure [Pa]		54	
Temperature [K]	297	473	473
Deposition time [min]	15	30	30

**Table 2 materials-15-04081-t002:** Roughness parameters of the pure Crofer 22APU and after individual stages of surface modification.

Samples	R_a_ [µm]	R_z_ [µm]	R_t_ [µm]
Crofer 22APU (unmodified)	0.35 ± 0.07	2.66 ± 0.24	4.32 ± 1.24
Crofer 22APU/Ar^+^	0.35 ± 0.02	2.05 ± 0.09	2.52 ± 0.34
Crofer 22APU/Ar^+^/SiC_x_N_y_:H	0.32 ± 0.01	3.12 ± 0.44	4.42 ± 1.27
Crofer 22APU/Ar^+^/N^+^/SiC_x_N_y_:H	0.31 ± 0.01	2.17 ± 0.24	3.28 ± 0.31

**Table 3 materials-15-04081-t003:** Ratio of half-areas covering the above-mentioned bands in FT IR spectra on substrates with a layer without N^+^ ion modification and prior N^+^ ion modification process.

Atomic Groups	Sic_x_N_y_:H and N^+^/Sic_x_N_y_:H Ratio	
	(001) Si	Crofer 22APU
Si-C	1.5	1.0
Si-C-N	0.6	1.0
Si-(CH_3_)n		0.6
Si-N	2.1	1.1
Si-CH_2_-Si	0.7	5.1
=NH	7.1	0.6
Si-CH_3_	1.75	0.8

**Table 4 materials-15-04081-t004:** Hardness (H_IT_) and Young’s modulus (E_IT_) for different indenter loads applied to the unmodified Crofer 22APU steel, Crofer 22APU steel with SiC_x_N_y_:H layer, and N^+^ ion modified Crofer 22APU steel with SiC_x_N_y_:H layer deposition.

Samples			H_IT_ [GPa]			E_IT_ [GPa]	
	Load [mN]	10	100	500	10	100	500
Crofer 22APU (unmodified)		3.1 ± 0.5	2.2 ± 0.1	1.7± 0.2	102 ± 6	123 ± 12	77 ± 5
Crofer 22APU/Ar^+^/SiC_x_N_y_:H		13.7 ± 1.5	4.1 ± 0.2	2.4 ± 0.1	135 ± 23	116 ± 5	100 ± 5
Crofer 22APU/Ar^+^/N^+^/SiC_x_N_y_:H		11.7 ± 1.0	3.5 ± 0.5	2.1 ± 0.1	200 ± 15	173 ± 29	78 ± 18

**Table 5 materials-15-04081-t005:** Wear rate and grove depth of unmodified Crofer 22APU steel and after plasma modification.

Sample	Load/Cycles	Groove Depth (d) [µm]	Wear Rate (WV) [mm^3^·N^−1^·m^−1^]
Crofer 22APU (unmodified)	1 N/2000	1.80 ± 0.26	51 ± 3
Crofer 22APU/Ar^+^/SiC_x_N_y_:H	1 N/1500	1.30 ± 0.13	71 ± 22
Crofer 22APU/Ar^+^/N^+^/SiC_x_N_y_:H	0.5 N/180	0.40 ± 0.14	503 ± 156

**Table 6 materials-15-04081-t006:** Parabolic rate constants of the studied layered systems obtained after 500 h of oxidation in air at 1073 K.

Tested Samples	t [h]	k_p_ [g^2^·cm^−4^·s^−1^]
Crofer 22APU (unmodified)	0–230 230–500	7.50 × 10^−14^ 1.02 × 10^−13^
Crofer 22APU/Ar^+^/SiC_x_N_y_:H	5–500	1.11 × 10^−13^
Crofer 22APU/Ar^+^/N^+^/SiC_x_N_y_:H	50–500	6.22 × 10^−14^

**Table 7 materials-15-04081-t007:** ASR and activation energy of electrical conduction determined for the studied layered systems after 500 h of oxidation in air at 1073 K.

Layered System	ASR [Ω·cm^2^]	E_c_ [eV]
Crofer 22APU (unmodified)/scale	0.0937	0.256 ± 0.007
Crofer 22APU/Ar^+^/SiC_x_N_y_:H/scale	0.0209	0.257 ± 0.006
Crofer 22APU/Ar^+^/N^+^/SiC_x_N_y_:H/scale	0.0101	0.241 ± 0.003

## Data Availability

The data presented in this study are available on request from the corresponding authors.

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
