# Peer review of "Plasmochemical Modification of Crofer 22APU for Intermediate-Temperature Solid Oxide Fuel Cell Interconnects Using RF PA CVD Method"

_materials, 2022, doi:10.3390/ma15124081_

Round 1

Reviewer 1 Report

In the article“Plasmochemical modification of Crofer 22APU for intermediate-temperature solid oxide fuel cell interconnects using RF PA CVD method”, a SiCxNy:H layer with (N+/SiCxNy:H series) or without the N+ ion modification process (SiCxNy:H series) were prepared on the Crofer 22APU ferritic stainless steel. Tests and analyses results are reasonably provided. For the benefit of the reader, a number of points need clarifying and certain statements require further justification. There are given below.

1) line 93: Fe-balance should be supplied in the composition (wt. %) of 22APU ferritic stainless steel.

2) why to choose the 1200-grit SiC abrasive papers to polished the samples? Is it reasonable to use only 1200# instead of the higher counts? It could be the cause of the delaminations in figure 1.

3) How about the thickness of each deposited layers?

4) Figure 4&5: lack scale

5) Line 147-151, unclear explanation on the change of the deposited layer (N+/SiCxNy:H) after 230 hrs. The author should give a better reason.

6) Figure 7: I think the BSE pattern should be used to display the different phases. Besides, where is the XRD result?

7) Figure 7: it seems that the statuses of the substrate are different.? How to distinguish the substrate and deposited layer?

8)Line 396-401: To identify different layers, the EDS or EDX result should be displayed based on figure 7. However, the author seem to omit the important data.

Author Response

Marta JanuÅ›, Ph. D., Eng. Kraków, 27.05.2022
AGH University of Science and Technology
Faculty of Materials Science and Ceramics
Department of Physical Chemistry and Modelling
A. Mickiewicza 30 Av., 30 059 Kraków, Poland
e-mail: martaj@agh.edu.pl
tel. +48 12 617 24 84, fax. +48 12 633 15 93
Dear Editor,
Find uploaded a copy of the manuscript entitled "Plasmochemical modification of Crofer 22 APU for intermediate-temperature solid oxide fuel cell interconnects using RF PA CVD method" written by Marta Januś, Karol Kyzioł, Stanisława Kluska, Witold Jastrzębski, Anna Adamczyk, Zbigniew Grzesik, Sławomir Zimowski, Marek Potoczek and Tomasz Brylewski.
We have revised the manuscript according to reviewer´s comments and the revisions in the manuscript have been indicated in red color.
We would like to thank You and the Reviewers once more for time and efforts in improving our manuscript.
On behalf of all authors,
Marta JanuÅ›

Reviewer 2 Report

Paper no. materials-1727877 titled Plasmochemical modification of Crofer 22APU for intermediate-temperature solid oxide fuel cell interconnects using RF PA CVD method by M. Janus et al.

This is an interesting paper that studies the performance of Crofer 22 APU for SOFC interconnects. The steel was coated with SiCxHy:N using radio-frequency plasma-activated chemical vapor deposition. The paper is publishable subject to revision.

1.There is no cross-section image of the deposited layer. Provide one to show the microstructure and layer thickness.

2.The authors say that they obtained amorphous silicon carbo—nitride layers (line 488). However, X-ray diffraction patterns are not provided. Please, confirm your conclusion by showing the XRD.

3.Scratch test images (Figs. 4, 5) have to include scale bars. The image of the bare steel should be included for the sake of comparison.

4.The parabolic rate constant of Crofer 22 APU should be compared with previous studies.

5.Authors say that they observed an amorphous silica scale (lines 410-411). However, SiO2 is not indicated in Fig. 7. Please, provide an EDS map to show the chemical composition of the oxide scale clearly.

6.Conclusions should be given point-by-point and numbered consecutively.

Author Response

(The authors gave the same response as above.)

Reviewer 3 Report

The authors have studied the effect of silicon carbonitride layer deposition on the surface of Crofer 22APU ferritic stainless steel on mechanical properties of the material and its resistance to high-temperature corrosion. Series of steel samples containing deposited layers with (N+/SiCxNy:H series) or without the N+ ion modification process (SiCxNy:H series) were investigated to estimate the usability of the materials as intermediate-temperature solid oxide fuel cell interconnects. The chemical composition, atomic structure, and microstructure were studied. Microhardness, Young’s modulus, wear rate, as well as electrical resistance were also determined. The authors obtained interesting results. The work is well done, but some deficiencies need to be corrected to make the manuscript acceptable for publication.

(1) Fig.7(b,c): Deposited layers should be indicated by arrows similarly as the corrosion products are indicated. Also please provide some XRD images for clarification.

(2) The term “interconnect/interconnector” should be unified.

(3) Lines 66–70: It is expected that among Ref. [10-14], to gain more recent references in the field, the following reference be added: https://doi.org/10.1016/j.compstruct.2021.114649

Author Response

(The authors gave the same response as above.)

Round 2

Reviewer 1 Report

All the issues have been well-treated, the paper can be accepted now.

Reviewer 2 Report

Authors answered my comments. The current version can be accepted for publication.